# Calibration Regularized Training of Deep Neural Networks using Dirichlet Kernel Density Estimation

## Abstract

Calibrated probabilistic classifiers are models whose predicted probabilities can directly be interpreted as uncertainty estimates. This property is particularly important in safety-critical applications such as medical diagnosis or autonomous driving. However, it has been shown recently that deep neural networks are poorly calibrated and tend to output overconfident predictions. As a remedy, we propose a trainable calibration error estimator based on Dirichlet kernel density estimates, which asymptotically converges to the true $L_p$ calibration error. This novel estimator enables us to achieve the strongest notion of multiclass calibration, called canonical calibration, while other common calibration methods only allow for top-label and marginal calibration. The empirical results show that our estimator is competitive with the state-of-the-art, consistently yielding tradeoffs between calibration error and accuracy that are (near) Pareto optimal across a range of network architectures. The computational complexity of our estimator is $\mathcal{O}(n^2)$, matching that of the kernel maximum mean discrepancy, used in a previously considered trainable calibration estimator (Kumar et al., 2018). By contrast, the proposed method has a natural choice of kernel, and can be used to generate consistent estimates of other quantities based on conditional expectation, such as the sharpness of an estimator.

## 1 Introduction

Deep neural networks have shown tremendous success in classification tasks, being regularly the best performing models in terms of accuracy. However, they are also known to make overconfident predictions (Guo et al., 2017), which is particularly problematic in safety-critical applications such as medical diagnosis or autonomous driving. Therefore, in many real world applications we do not just care about the predictive performance, but also about the trustworthiness of that prediction, that is, we are interested in accurate predictions with robust uncertainty estimates. To this end, we want our models to be uncertainty calibrated which means that, for instance, among all cells that have been predicted with a probability of 0.8 to be cancerous, in fact a fraction of 80 % belong to a malignant tumor.

Being calibrated, however, does not imply that the classifier achieves good accuracy. For instance, a classifier that always predicts the marginal distribution of the target class is calibrated, but will not be very useful in practice. Likewise, a good predictive performance does not ensure calibration. In particular, for a broad class of loss functions, risk minimization leads to asymptotically Bayes optimal classifiers (Bartlett et al., 2006). However, there is no guarantee that they are calibrated, even in the aysmptotic limit. Therefore, we consider minimizing the risk plus a term that penalizes miscalibration, i.e., $\mathrm{Risk} + \lambda \cdot \mathrm{CalibrationError}$. For parameter values $\lambda > 0$, this will push the classifier towards a calibrated model, while maintaining similar accuracy. The existence of such a $\lambda > 0$ is suggested by the fact that there always exists at least one Bayes optimal classifier that is calibrated, namely $\mathbb{P}(y|x)$.

To optimize the risk and the calibration error jointly, we propose a differentiable and consistent estimator of the expected $L_p$ calibration error based on kernel density estimates (KDEs). In particular, we use a Beta kernel in binary classification tasks and a Dirichlet kernel in the multiclass setting,

as these kernels are the natural choices to model density estimation over a probability simplex. Our Dirichlet kernel based estimator allows for the estimation of canonical calibration, which is the strongest notion of multiclass calibration as it implies the calibration of the whole probability vector (Bröcker, 2009; Appice et al., 2015; Vaicenavicius et al., 2019). By contrast, most other state-of-the-art methods only achieve weaker versions of multiclass calibration, namely top-label (Guo et al., 2017) and marginal or class-wise calibration (Kull et al., 2019). The top-label calibration only considers the scores for the predicted class, while for marginal calibration the multiclass problem is split up into $K$ one-vs-all binary ones, each of which is required to be calibrated according to the definition of binary calibration. In many applications marginal and canonical calibration are preferable to top-label calibration, since we often care about having reliable uncertainty estimates for more than just one class per prediction. For instance, in medical diagnosis we do not just care about the most likely disease a certain patient might have but also about the probabilities of other diseases.

Our contributions can be summarized as follows:

1. We develop a trainable calibration error objective using Dirichlet kernel density estimates, which can be minimized alongside any loss function in the existing batch stochastic gradient descent framework.
2. We propose to use our estimator to evaluate *canonical calibration*. Due to the scaling properties of Dirichlet kernel density estimation, and the tendency for probabilities to be concentrated in a relatively small number of classes, this becomes feasible in cases that cannot be estimated using a binned estimator.
3. We show on a variety of network architectures and two datasets that DNNs trained alongside an estimator of the calibration error achieve competitive results both on existing metrics and on the proposed measure of canonical calibration.

## 2 RELATED WORK

Calibration of probabilistic predictors has long been studied in many fields. This topic gained attention in the deep learning community following the observation in Guo et al. (2017) that modern neural networks are poorly calibrated and tend to give overconfident predictions due to overfitting on the NLL loss. The surge of interest resulted in many calibration strategies that can be split in two general categories, which we discuss subsequently. **Post-hoc calibration strategies** learn a calibration map of the predictions from a trained predictor in a post-hoc manner. For instance, Platt scaling (Platt, 1999) fits a logistic regression model on top of the logit outputs of the model. A special case of Platt scaling that fits a single scalar, called temperature, has been popularized by Guo et al. (2017) as an accuracy-preserving, easy to implement and effective method to improve calibration. However, it has the undesired consequence that it clamps the high confidence scores of accurate predictions (Kumar et al., 2018). Other approaches for post-hoc calibration include: histogram binning (Zadrozny & Elkan, 2001), isotonic regression (Zadrozny & Elkan, 2002), and Bayesian binning into quantiles (Naeini & Cooper, 2015). **Trainable calibration strategies** integrate a differentiable calibration measure into the training objective. One of the earliest approaches is regularization by penalizing low entropy predictions (Pereyra et al., 2017). Similarly to temperature scaling, it has been shown that entropy regularization needlessly suppresses high confidence scores of correct predictions (Kumar et al., 2018). Another popular strategy is MMCE (Maxmimum Mean Calibration Error) (Kumar et al., 2018), where the entropy regularizer is replaced by a kernel-based surrogate for the calibration error that can be optimized alongside NLL. It has been shown that label smoothing (Szegedy et al., 2015; Müller et al., 2020), i.e. training models with a weighted mixture of the labels instead of one-hot vectors, also improves model calibration. Liang et al. (2020) propose to add the difference between predicted confidence and accuracy as auxiliary term to the cross-entropy loss. Focal loss (Mukhoti et al., 2020; Lin et al., 2018) has recently been *empirically* shown to produce better calibrated models than many of the alternatives, but does not estimate a clear quantity related to calibration error.

**Kernel density estimation** (Parzen, 1962; Rosenblatt, 1956) is a non-parametric method to estimate a probability density function from a finite sample. Zhang et al. (2020) propose a KDE-based estimator of the calibration error for measuring calibration performance. However, they use the triweight kernel, which has a limited support interval and is therefore applicable to binary classification, but does not have a natural extension to higher dimensional simplexes, in contrast to the Dirichlet kernel

that we consider here. As a result, they consider an unnatural proxy to marginal calibration error, which does not result in a consistent estimator.

## 3 METHODS

The most commonly used loss functions are designed to achieve consistency in the sense of Bayes optimality under risk minimization, however, they do not guarantee calibration - neither for finite samples nor in the asymptotic limit. Since we are interested in models $f$ that are both accurate and calibrated, we consider the following optimization problem bounding the calibration error $\mathrm{CE}(f)$:

$$f = \arg\min_{f \in \mathcal{F}} \mathrm{Risk}(f), \text{s.t. } \mathrm{CE}(f) \leq B \tag{1}$$

for some $B > 0$, and its associated Lagrangian

$$f = \arg\min_{f \in \mathcal{F}} \Big( \mathrm{Risk}(f) + \lambda \cdot \mathrm{CE}(f) \Big). \tag{2}$$

We measure the (mis-)calibration in terms of the $L_p$ calibration error. To this end, let $(\Omega, \mathcal{A}, \mathbb{P})$ be a probability space, let $\mathcal{X} = \mathbb{R}^d, \mathcal{Y} = \{0, 1, ..., K\}$. Let $x : \Omega \to \mathcal{X}$ and $y : \Omega \to \mathcal{Y}$ be random variables while realizations are denoted with subscripts. Furthermore, let $f : \mathcal{X} \to \triangle^K$ be a decision function, where $\triangle^K$ denotes the $K$ dimensional simplex as is achieved e.g. from the output of a final softmax layer in a neural network.

**Definition 3.1** (Calibration error, (Naeini et al., 2015; Kumar et al., 2019; Wenger et al., 2020)). *The $L_p$ calibration error of $f$ is:*

$$\mathrm{CE}_p(f) = \left( \mathbb{E}\left[ \left\| \mathbb{E}[y \mid f(x)] - f(x) \right\|_p^p \right] \right)^{\frac{1}{p}}. \tag{3}$$

We note that we consider multiclass calibration, and that $f(x)$ and the conditional expectation in Equation 3 therefore map to points on a probability simplex. We say that a classifier $f$ is perfectly calibrated if $\mathrm{CE}_p(f) = 0$. Kumar et al. (2018) have also considered a minimization problem similar to Equation 2. Instead of using the $\mathrm{CE}_p$ they use a metric called maximum mean calibration error (MMCE) that is 0 if and only if $\mathrm{CE}_p = 0$. However, it is unclear how MMCE relates to the canonical multiclass setting or to the norm parameter $p$ for non-zero $\mathrm{CE}_p$.

In order to optimize Definition 3.1 directly, we need to perform density estimation over the probability simplex in order to empirically compute the conditional expectation. In a binary setting, this has traditionally been done with binned estimates (Naeini et al., 2015; Guo et al., 2017; Kumar et al., 2019). However, this is not differentiable w.r.t. the function $f$, and cannot be incorporated into a gradient based training procedure. Furthermore, binned estimates suffer from the curse of dimensionality and do not have a practical extension to multiclass settings. A natural choice for a differentiable kernel density estimator in the binary case is a kernel based on the Beta distribution and the extension to the multiclass case is given by the Dirichlet distribution. Hence, we consider an estimator for the $\mathrm{CE}_p$ based on Beta and Dirichlet kernel density estimates in the binary and multiclass setting, respectively. We require that this estimator is consistent and differentiable such that we can train it according to Equation 2. This estimator is given by:

$$\widehat{\mathrm{CE}_p(f)}^p = \frac{1}{n} \sum_{h=1}^n \left[ \left\| \mathbb{E}[\widehat{y \mid f(x)}] \Big|_{f(x_h)} - f(x_h) \right\|_p^p \right], \tag{4}$$

where $\mathbb{E}[\widehat{y \mid f(x)}] \Big|_{f(x_h)}$ denotes $\mathbb{E}[\widehat{y \mid f(x)}]$ evaluated at $f(x) = f(x_h)$. If $\mathbb{P}_{x,y}$ has a probability density $p_{x,y}$ with respect to the product of the Lebesgue and counting measure, we can define: $p_{x,y}(x_i, y_i) = p_{y|x=x_i}(y_i) \, p_x(x_i)$. Then we define the estimator of the conditional expectation as follows:

$$\mathbb{E}[y \mid f(x)] = \sum_{y_k \in \mathcal{Y}} y_k \, p_{y|x=f(x)}(y_k) = \frac{\sum_{y_k \in \mathcal{Y}} y_k \, p_{x,y}(f(x), y_k)}{p_x(f(x))} \tag{5}$$

$$\approx \frac{\sum_{i=1}^n k(f(x); f(x_i)) y_i}{\sum_{i=1}^n k(f(x); f(x_i))} =: \mathbb{E}[\widehat{y \mid f(x)}] \tag{6}$$

where $k$ is the kernel of a kernel density estimate evaluated at point $x_i$.

**Proposition 3.2.** $\widehat{\mathbb{E}[y \mid f(x)]}$ *is a pointwise consistent estimator of* $\mathbb{E}[y \mid f(x)]$, *that is:*

$$\lim_{n \to \infty} \frac{\sum_{i=1}^{n} k(f(x); f(x_i)) y_i}{\sum_{i=1}^{n} k(f(x); f(x_i))} = \frac{\sum_{y_k \in \mathcal{Y}} y_k \, p_{x,y}(f(x), y_k)}{p_x(f(x))}. \tag{7}$$

*Proof.* By the consistency of kernel density estimators (Silverman, 1986; Chen, 1999; Ouimet & Tolosana-Delgado, 2021), for all $f(x) \in (0,1)$, $\frac{1}{n} \sum_{i=1}^{n} k(f(x); f(x_i)) y_i \xrightarrow{n \to \infty}$ $\sum_{y_k \in \mathcal{Y}} y_k \, p_{x,y}(f(x), y_k)$ and $\frac{1}{n} \sum_{i=1}^{n} k(f(x); f(x_i)) \xrightarrow{n \to \infty} p_x(f(x))$. The fact that the ratio of two convergent sequences converges against the ratio of their limits shows the result. $\square$

**Mean squared error in binary classification** As a first instantiation of our framework we consider a binary classification setting, with the mean squared error $\mathrm{MSE}(f) = \mathbb{E}[(f(x) - y)^2]$ as the risk function, jointly optimized with the $L_2$ calibration error $\mathrm{CE}_2$. Following Murphy (1973); Degroot & Fienberg (1983); Kuleshov & Liang (2015); Nguyen & O'Connor (2015) we decompose (full derivation in Appendix A) the MSE as:

$$\mathrm{MSE}(f) - \mathrm{CE}_2(f)^2 = \mathbb{E}\left[ \left(1 - \mathbb{E}[y \mid f(x)]\right) \mathbb{E}[y \mid f(x)] \right] \geq 0. \tag{8}$$

Similar to Equation 2, we consider the optimization problem for some $\lambda > 0$:

$$f = \arg\min_{f \in \mathcal{F}} \left( \mathrm{MSE}(f) + \lambda \, \mathrm{CE}_2(f)^2 \right). \tag{9}$$

Using Equation 8 we rewrite:

$$\mathrm{MSE}(f) + \lambda \, \mathrm{CE}_2(f)^2 = (1 + \lambda) \, \mathrm{MSE}(f) - \lambda \left( \mathrm{MSE}(f) - \mathrm{CE}_2(f)^2 \right) \tag{10}$$

$$= (1 + \lambda) \, \mathrm{MSE}(f) - \lambda \mathbb{E}\left[ \left(1 - \mathbb{E}[y \mid f(x)]\right) \mathbb{E}[y \mid f(x)] \right]. \tag{11}$$

Rescaling Equation 11 by a factor of $(1 + \lambda)^{-1}$ and a variable substitution $\gamma = \frac{\lambda}{1+\lambda} \in [0, 1)$

$$f = \arg\min_{f \in \mathcal{F}} \left( \mathrm{MSE}(f) + \lambda \, \mathrm{CE}_2(f)^2 \right) = \arg\min_{f \in \mathcal{F}} \left( \mathrm{MSE}(f) - \gamma \mathbb{E}\left[ \left(1 - \mathbb{E}[y \mid f(x)]\right) \mathbb{E}[y \mid f(x)] \right] \right) \tag{12}$$

$$= \arg\min_{f \in \mathcal{F}} \left( \mathrm{MSE}(f) + \gamma \mathbb{E}\left[ \mathbb{E}[y \mid f(x)]^2 \right] \right). \tag{13}$$

For optimization we wish to find an estimator for $\mathbb{E}[\mathbb{E}[y \mid f(x)]^2]$. Building upon Equation 6, a partially debiased estimator can be written as:[1]

$$\mathbb{E}\left[ \widehat{\mathbb{E}[y \mid f(x)]^2} \right] \approx \frac{1}{n} \sum_{h=1}^{n} \frac{\left( \sum_{i \neq h} k(f(x_h); f(x_i)) y_i \right)^2 - \sum_{i \neq h} \left( k(f(x_h); f(x_i)) y_i \right)^2}{\left( \sum_{i \neq h} k(f(x_h); f(x_i)) \right)^2 - \sum_{i \neq h} \left( k(f(x_h); f(x_i)) \right)^2}. \tag{14}$$

In a binary setting, the kernels $k(\cdot, \cdot)$ are Beta distributions, i.e. denoting $z_i := f(x_i)$ for short, then:

$$k_{\mathrm{Beta}}(z, z_i) := z^{\alpha_i - 1} (1 - z)^{\beta_i - 1} \frac{\Gamma(\alpha_i + \beta_i)}{\Gamma(\alpha_i) \, \Gamma(\beta_i)}, \tag{15}$$

with $\alpha_i = \frac{z_i}{h} + 1$ and $\beta_i = \frac{1 - z_i}{h} + 1$ (Chen, 1999; Bouezmarni & Rolin, 2003; Zhang & Karunamuni, 2010), where $h$ is a bandwidth parameter in the kernel density estimate that goes to 0 as $n \to \infty$. We note that the computational complexity of this estimator is $\mathcal{O}(n^2)$. Within the gradient descent training procedure, the density is estimated using a mini-batch and therefore the $\mathcal{O}(n^2)$ complexity is w.r.t. a mini-batch, not the entire dataset.

The estimator in Equation 14 is a ratio of two second order U-statistics that converge as $n^{-1/2}$ (Ferguson, 2005). Therefore, the overall convergence will be $n^{-1/2}$. Empirical covergence rates are calculated in Appendix D.3 and shown to be close to the theoretically expected value.

---

[1]We have debiased the numerator and denominator individually (Ferguson, 2005, Section 2), but for simplicity have not corrected for the fact that we are estimating a ratio (Scott & Wu, 1981).

**Multiclass calibration with Dirichlet kernel density estimates** There are multiple definitions regarding multiclass calibration that differ in the strictness regarding the calibration of the probability vector $f(x)$. The weakest notion is top label calibration, which, as the name suggests, only cares about calibrating the entry with the highest predicted probability, which reduces to a binary calibration problem again (Guo et al., 2017). Marginal or class-wise calibration (Kull et al., 2019) is the most commonly used definition of multiclass calibration and a stronger version of top label calibration. Here, the problem is split into $K$ one-vs-all binary calibration setting, such that each class has to be calibrated against the other $K - 1$ classes:

$$\text{MCE}_p(f)^p = \sum_{k=1}^{K} \mathbb{E}\left[\left|\mathbb{E}[y = k \mid f(x)_k] - f(x)_k\right|^p\right]. \tag{16}$$

An estimator for this calibration error is:

$$\widehat{\text{MCE}_p(f)}^p = \sum_{k=1}^{K} \frac{1}{n} \sum_{j=1}^{n} \left| \frac{\sum_{i \neq j} k_{\text{Beta}}(f(x_j)_k; f(x_i)_k)[y_i]_k}{\sum_{i \neq j} k_{\text{Beta}}(f(x_j)_k; f(x_i)_k)} - f(x_j)_k \right|^p. \tag{17}$$

The strongest notion of multiclass calibration, and the one that we want to consider in this paper, is called canonical calibration (Bröcker, 2009; Appice et al., 2015; Vaicenavicius et al., 2019). Here it is required that the whole probability vector $f(x)$ is calibrated. The definition is exactly the one from Definition 3.1. Its estimator is:

$$\widehat{\text{CE}_p(f)}^p = \frac{1}{n} \sum_{j=1}^{n} \left\| \frac{\sum_{i \neq j} k_{\text{Dir}}(f(x_j); f(x_i))y_i}{\sum_{i \neq j} k_{\text{Dir}}(f(x_j); f(x_i))} - f(x_j) \right\|_p^p \tag{18}$$

where $k_{\text{Dir}}$ is a Dirichlet kernel defined as:

$$k_{\text{Dir}}(z, z_i) := \frac{\Gamma(\sum_{i=1}^{K} \alpha_i)}{\prod_{i=1}^{K} \Gamma(\alpha_i)} \prod_{j=1}^{K} z_j^{\alpha_{ij} - 1} \tag{19}$$

with $\alpha_i = z_i/h + 1$ (Ouimet & Tolosana-Delgado, 2021). As before, the computational complexity is $\mathcal{O}(n^2)$ irrespective of $p$.

This estimator is differentiable and furthermore, the following proposition holds:

**Proposition 3.3.** *The Dirichlet kernel based* CE *estimator is consistent, that is*

$$\lim_{n \to \infty} \frac{1}{n} \sum_{j=1}^{n} \left\| \frac{\sum_{i \neq j}^{n} k_{\text{Dir}}(f(x_j); f(x_i))y_i}{\sum_{i \neq j}^{n} k_{\text{Dir}}(f(x_j); f(x_i))} - f(x_j) \right\|_p^p = \mathbb{E}\left[\left\|\mathbb{E}[y \mid f(x)] - f(x)\right\|_p^p\right]. \tag{20}$$

*Proof.* Dirichlet kernel estimators are consistent (Ouimet & Tolosana-Delgado, 2021), consequently, by Proposition 3.2 the term inside the norm is consistent for any fixed $f(x_j)$ (note, that summing over $i \neq j$ ensures that the ratio of the KDE's does not depend on the outer summation). Moreover, for any convergent sequence also the norm of that sequence converges against the norm of its limit. Ultimately, the outer sum is merely the sample mean of consistent summands, which again is consistent. □

## 4 EMPIRICAL SETUP

We trained ResNet (He et al., 2015), ResNet with stochastic depth (SD) (Huang et al., 2016), DenseNet (Huang et al., 2018) and WideResNet (Zagoruyko & Komodakis, 2016) networks on CIFAR-10 and CIFAR-100 (Krizhevsky, 2009). We use 45000 images for training. The code will be released upon acceptance.

**Baselines** *Cross-entropy*: The first baseline model is trained using cross-entropy with the data preprocessing, training procedure and hyperparameters described in the corresponding paper for the architecture. *Trainable calibration strategies* **MMCE** (Kumar et al., 2018) is a differentiable measure of calibration with a property that it is minimized at perfect calibration. It is used as a regulariser alongside NLL, with the strength of regularization parameterized by $\lambda$. **Focal loss** (Mukhoti et al., 2020) is an alternative to the popular cross-entropy loss, defined as $\mathcal{L}_f = -(1 -$

$f(y|x))^\gamma \log(f(y|x))$, where $\gamma$ is a hyperparameter and $f(y|x)$ is the probability score that a neural network $f$ outputs for a class $y$ on an input $x$. Their best-performing approach is the sample-dependent FL-53 where $\gamma = 5$ for $f(y|x) \in [0, 0.2)$ and $\gamma = 3$ otherwise, followed by the method with fixed $\gamma = 3$. *Post-hoc calibration strategies* Guo et al. (2017) investigated the performance of several post-hoc calibration methods and found **temperature scaling** to be a strong baseline, which we use as a representative of this group. It works by scaling the logits with a scalar $T > 0$, typically learned on a validation set by minimizing NLL. Following Kumar et al. (2018); Mukhoti et al. (2020), we also use temperature scaling as a post-processing step for our method.

**Metrics**   The most widely-used metric for expected calibration error (ECE) is a binned estimator (Naeini et al., 2015), which divides the interval $[0, 1]$ into bins of equal width and then calculates a weighted average of the absolute difference between accuracy and confidence for each bin. A better binning scheme involves determining the bin sizes so that an equal number of samples fall into each bin (Nguyen & O'Connor, 2015; Mukhoti et al., 2020). We report the ECE (%) with 15 bins calculated according to the latter, so-called adaptive binning procedure. We compute the 95% confidence intervals using 100 bootstrap samples as in Kumar et al. (2019). We consider multiple versions of the ECE metric based on the $L_p$ norm and the type of calibration (top-label, marginal, canonical). Top-label calibration error only considers the probability of the predicted class, marginal requires per-class calibration and the canonical is the highest form of calibration which requires the entire probability vector to be calibrated. We report $L_1$ and $L_2$ ECE in the marginal and canonical case. Additional experiments with top-label and marginal calibration on both CIFAR-10 and CIFAR-100 can be found in Appendix B.

**Hyperparameters**   A crucial parameter for KDE is the bandwidth, a positive number that defines the smoothness of the density plot. Poorly chosen bandwidth may lead to undersmoothing (small bandwidth) or oversmoothing (large bandwidth). A commonly used non-parametric bandwidth selector is maximum likelihood cross validation (Duin, 1976). For our experiments we choose the bandwidth from a list of possible values by maximizing the leave-one-out likelihood. The $\lambda$ parameter for weighting the calibration error w.r.t the loss is typically chosen via cross-validation or using a holdout validation set. The $p$ parameter is chosen depending on the desired $L_p$ calibration error and the corresponding theoretical guarantees.

## 5   RESULTS AND DISCUSSION

### 5.1   BINARY CLASSIFICATION

We construct a binary experiment by splitting the CIFAR-10 classes into 2 classes: vehicles (plane, automobile, ship, truck) and animals (bird, cat, deer, dog, frog, horse). Figure 1a shows how the choice of the bandwidth parameter influences the shape of the estimate.

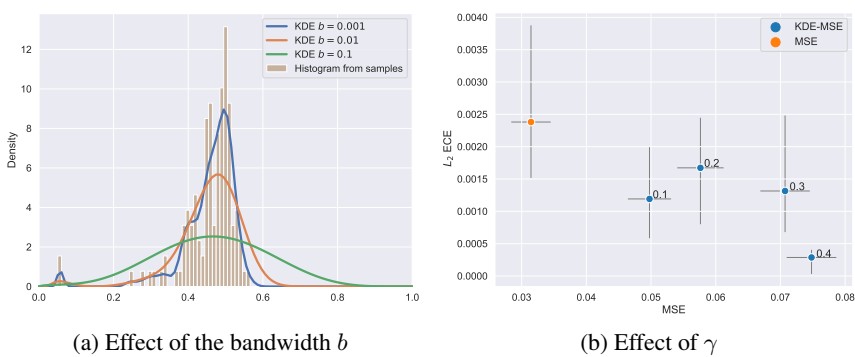

(a) Effect of the bandwidth $b$          (b) Effect of $\gamma$

Figure 1: Calibration regularized training using MSE loss and $CE_2$

Figure 1b shows the effect of the regularization parameter $\gamma$ on the performance of a ResNet-110 model. The orange point represents a model trained with MSE loss, and the blue points (KDE-MSE) correspond to models trained with regularized MSE loss by an $L_2$ calibration error for different values of $\gamma$. As expected, the calibration regularized training decreases the $L_2$ calibration error at the cost of slightly increased error.

## 5.2 EVALUATING CANONICAL CALIBRATION

Accurately evaluating the calibration error is another crucial step towards designing trustworthy models that can be used in high-cost settings. In spite of its numerous flaws discussed in Vaicenavicius et al. (2019); Ding et al. (2020); Ashukha et al. (2021), such as its sensitivity to the binning scheme, the histogram-based estimator remains the most widely used metric for evaluating miscalibration. Another downside of the binned estimator is its inability to capture canonical calibration due to the curse of dimensionality, as the number of bins grows exponentially with the number of classes. Therefore, because of its favourable scaling properties, we propose using our Dirichlet kernel density estimate as an alternative metric (KDE-ECE) to measure calibration.

To investigate its relationship with the commonly used binned estimator, we first introduce an extension of the top-label binned estimator to the probability simplex in the three class setting. We start by partitioning the probability simplex into equally-sized, triangle-shaped bins and assign the probability scores to the corresponding bin, as shown in Figure 2a. Then, we define the binned estimate of canonical calibration error as follows:

$$\mathrm{CE}_p(f)^p \approx \mathbb{E}\left[\|H(f(x)) - f(x)\|_p^p\right] \approx \frac{1}{n}\sum_{i=1}^{n}\|H(f(x_j)) - f(x_i)\|_p^p \qquad (21)$$

where $H(f(x_j))$ is the histogram estimate, shown in Figure 2b. The surface of the corresponding Dirichlet KDE is presented in Figure 2c. In Figure 3 we show that the KDE-ECE estimates of the three types of calibration closely correspond to the their histogram-based approximations. Each point in the plot represents a ResNet-56 model trained on a different subset of three classes from CIFAR-10. See Appendix C for another example of the binned estimator and Dirichlet KDE on CIFAR-10 and an experiment with varying number of points used for the density estimation.

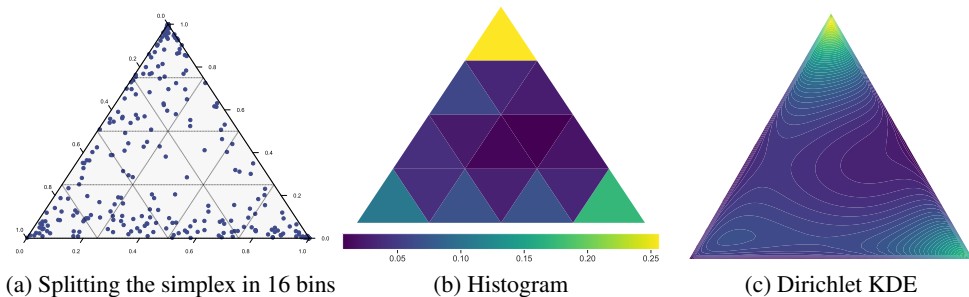

  (a) Splitting the simplex in 16 bins     (b) Histogram      (c) Dirichlet KDE

Figure 2: Extension of the binned estimator to the probability simplex, compared with the KDE-ECE. The KDE-ECE achieves a better approximation to the finite sample, and accurately models the fact that samples tend to be concentrated near low dimensional faces of the simplex.

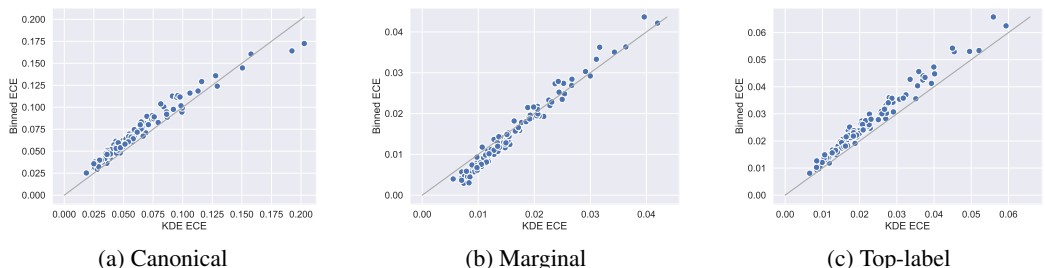

    (a) Canonical         (b) Marginal         (c) Top-label

Figure 3: Relationship between the KDE-ECE estimates and their corresponding binned approximations on the three types of calibration. Each point represents a ResNet-56 model trained on a subset of three classes from CIFAR-10. The 3000 probability scores of the test set are assigned in 25 bins with adaptive width for the binned estimate. A bandwidth of 0.001 is used for KDE-ECE.

### 5.3 MULTICLASS CLASSIFICATION

In this section we evaluate our proposed KDE-based ECE estimator that was jointly trained with cross entropy loss (KDE-CRE) against other baselines in a multiclass setting on CIFAR-10 and CIFAR-100. We found that for KDE-CRE, values of $\lambda \in [0.01, 0.1]$ provide a good trade-off in terms of accuracy and calibration error. Table 1 summarizes the accuracy and marginal $L_1$ ECE% (computed using 15 bins), measured across multiple architectures. For MMCE, we report the results with $\lambda = 1$ and for KDE-CRE we use $\lambda = 0.01$. An analogous table measuring marginal $L_2$ ECE is given in Appendix B.

Table 1: Accuracy and marginal $L_1$ ECE (%) computed with 15 bins for different loss functions and architectures, both trained from scratch (Pre T) and after temperature scaling on a validation set (Post T). Best results are marked in bold.

| Loss | Metric | | CIFAR-10 | | | | CIFAR-100 | | | |
| --- | --- | --- | --- | --- | --- | --- | --- | --- | --- | --- |
| | | | ResNet | ResNet (SD) | Wide-ResNet | DenseNet | ResNet | ResNet (SD) | Wide-ResNet | DenseNet |
| CRE | ECE | Pre T | 0.419 | 0.357 | **0.241** | 0.236 | 0.129 | 0.100 | **0.086** | **0.090** |
| | | Post T | 0.282 | 0.250 | 0.278 | **0.165** | 0.114 | **0.089** | **0.105** | **0.078** |
| | Acc | Pre T | 0.925 | **0.926** | **0.957** | 0.947 | **0.700** | **0.728** | **0.803** | 0.756 |
| | | Post T | **0.927** | 0.925 | **0.957** | 0.947 | **0.700** | **0.729** | **0.801** | 0.758 |
| MMCE | ECE | Pre T | **0.250** | 0.390 | 0.265 | **0.193** | 0.143 | 0.100 | 0.120 | 0.123 |
| | | Post T | 0.361 | 0.308 | 0.291 | 0.235 | 0.121 | 0.093 | 0.109 | 0.124 |
| | Acc | Pre T | **0.929** | 0.925 | 0.947 | 0.944 | 0.693 | 0.723 | 0.767 | 0.748 |
| | | Post T | 0.926 | **0.926** | 0.949 | 0.945 | 0.691 | 0.722 | 0.770 | 0.743 |
| FL-53 | ECE | Pre T | 0.403 | 0.416 | 0.414 | 0.259 | 0.145 | 0.120 | 0.125 | 0.095 |
| | | Post T | 0.272 | 0.267 | 0.437 | 0.220 | 0.124 | 0.107 | 0.106 | 0.081 |
| | Acc | Pre T | 0.922 | 0.920 | 0.936 | **0.948** | 0.695 | 0.711 | 0.760 | 0.752 |
| | | Post T | 0.923 | 0.919 | 0.936 | **0.949** | 0.693 | 0.712 | 0.763 | 0.753 |
| $L_1$ **KDE-CRE** | ECE | Pre T | 0.363 | **0.338** | 0.289 | 0.296 | **0.128** | **0.096** | 0.092 | 0.099 |
| | | Post T | **0.182** | **0.220** | **0.226** | 0.248 | **0.104** | 0.095 | 0.108 | 0.085 |
| | Acc | Pre T | 0.926 | 0.925 | 0.953 | 0.943 | 0.697 | 0.725 | 0.796 | **0.757** |
| | | Post T | **0.927** | 0.925 | 0.953 | 0.944 | 0.698 | 0.720 | 0.793 | **0.759** |

We notice that for both pre and post temperature scaling, KDE-CRE achieves very competitive ECE scores. Another encouraging observation is that the improvement of calibration error comes at almost no cost in accuracy. An important advantage of our KDE-based method is the ability to directly train and evaluate canonical calibration. In Figure 4 we show a scatter plot with confidence intervals of the $L_1$ and $L_2$ KDE-CRE models for canonical calibration and the other baselines on CIFAR-10. We measure the canonical calibration using our KDE-ECE metric from section 5.2. In three of the architectures, both $L_1$ and $L_2$ KDE-CRE either dominate or are statistically tied with cross-entropy (CRE). Similarly, Figure 5 shows a scatter plot of $L_1$ and $L_2$ KDE-CRE models trained to minimize marginal calibration error. In this case, we measure $L_2$ marginal ECE with the standard binned estimator. In most cases, our methods Pareto dominate the other baselines. A general observation can be made, however, that the models trained with cross-entropy have a surprisingly low marginal calibration error, contrary to previous findings that show poor calibration when considering only the most confident prediction (top-label calibration). An additional experiment comparing the CRE baseline with KDE-CRE for canonical calibration on a benchmark dataset of histological images of human colorectal cancer is given in Appendix D.2, which clearly illustrates the superior performance of our method, both in terms of accuracy and calibration error in this context.

To summarize, the experiments show that our estimator is consistently producing competitive calibration errors with other state-of-the-art approaches, while maintaining accuracy and keeping the computational complexity at $\mathcal{O}(n^2)$. We evaluate the computational overhead of CRE and KDE-CRE and summarize the results in a table in Appendix D.1, which shows that the added cost is less than a couple percent. There are several limitations in the current work: A larger scale benchmarking will be beneficial for exploring the limits of canonical calibration using Dirichlet kernels. Furthermore, while we showed consistency of our estimator, we did not fully derive and implement its debiasing. Due to space constraints, this was not the focus of the paper and is left for future work.

## 6 CONCLUSION

In this paper, we proposed a consistent and differentiable estimator of an $L_p$ calibration error using Dirichlet kernels. The KDE-based estimate can be directly optimized alongside any loss function in the existing batch stochastic gradient descent framework. Furthermore, we propose using it as a mea-

sure of the highest form of calibration which requires the entire probability vector to be calibrated. We showed empirically on a range of neural architectures that the performance of our estimator in terms of accuracy and calibration error is competitive against the current state-of-the-art, while having superior properties as a consistent estimator of canonical calibration error.

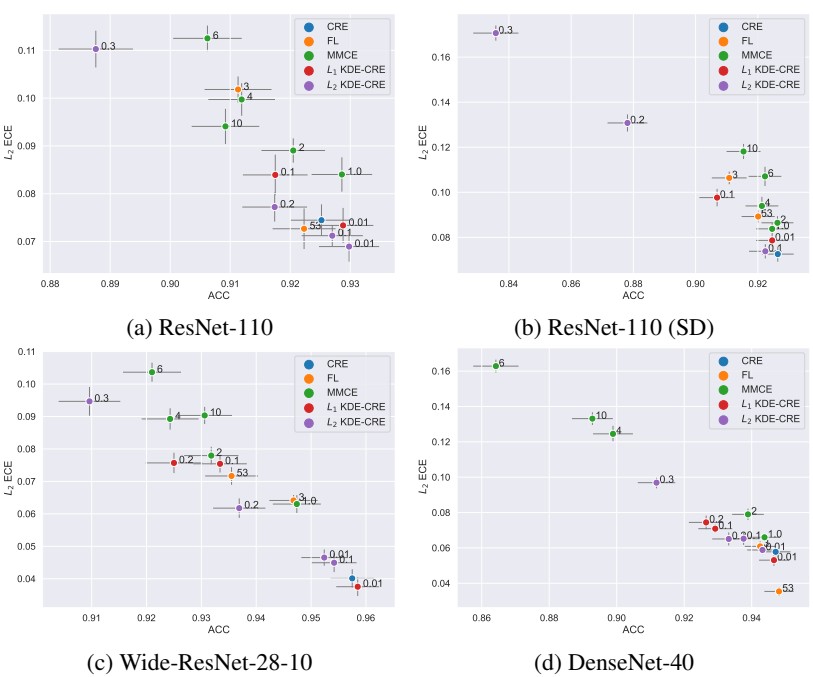

Figure 4: Canonical calibration on CIFAR-10

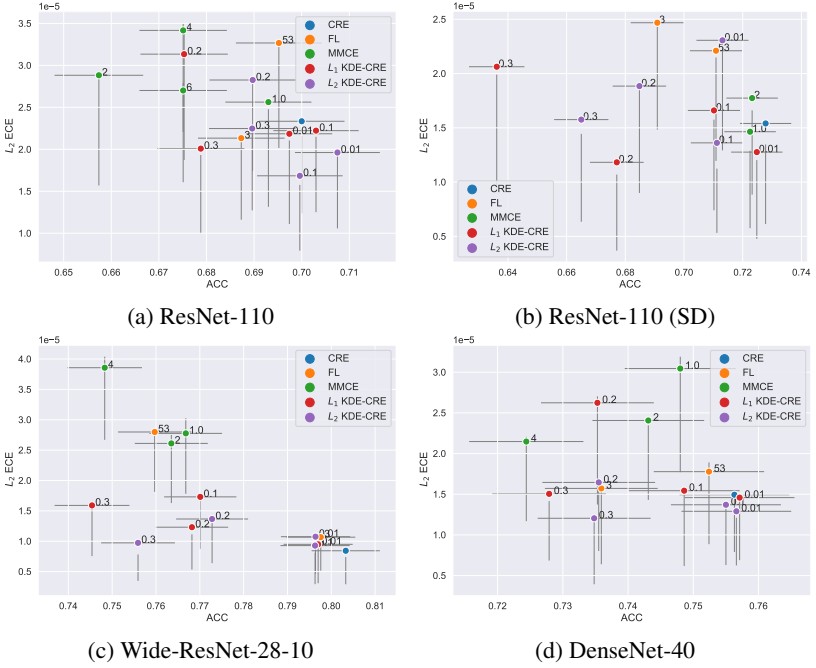

Figure 5: Marginal calibration on CIFAR-100

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

## A  DERIVATION OF THE MSE DECOMPOSITION

**Definition A.1** (Mean Squared Error (MSE))**.** *The mean squared error of an estimator is*

$$\mathrm{MSE}(f) := \mathbb{E}[(f(x) - y)^2]. \tag{22}$$

**Proposition A.2.** $\mathrm{MSE}(f) \geq \mathrm{CE}_2(f)^2$

*Proof.*

$$\mathrm{MSE}(f) := \mathbb{E}[(f(x) - y))^2] = \mathbb{E}[((f(x) - \mathbb{E}[y \mid f(x)]) + (\mathbb{E}[y \mid f(x)] - y))^2] \tag{23}$$

$$= \underbrace{\mathbb{E}[(f(x) - \mathbb{E}[y \mid f(x)])^2]}_{=CE_2^2} + \mathbb{E}[(\mathbb{E}[y \mid f(x)] - y)^2] \tag{24}$$

$$+ 2\mathbb{E}[(f(x) - \mathbb{E}[y \mid f(x)])(\mathbb{E}[y \mid f(x)] - y)]$$

which implies

$$\mathrm{MSE}(f) - \mathrm{CE}_2(f)^2 = \mathbb{E}[(\mathbb{E}[y \mid f(x)] - y)^2] \tag{25}$$

$$+ 2\mathbb{E}[(f(x) - \mathbb{E}[y \mid f(x)])(\mathbb{E}[y \mid f(x)] - y)]$$

$$= \mathbb{E}[(\mathbb{E}[y \mid f(x)] - y)^2] + 2\mathbb{E}[(f(x)\mathbb{E}[y \mid f(x)]] \tag{26}$$

$$- 2\mathbb{E}[f(x)y] - 2\mathbb{E}[\mathbb{E}[y \mid f(x)]^2] + 2\mathbb{E}[\mathbb{E}[y \mid f(x)]y]]$$

$$= \mathbb{E}[\mathbb{E}[y \mid f(x)]^2] + \mathbb{E}[y^2] - 2\mathbb{E}[\mathbb{E}[y \mid f(x)]y] \tag{27}$$

$$+ 2\mathbb{E}[(f(x)\mathbb{E}[y \mid f(x)]] - 2\mathbb{E}[f(x)y]$$

$$- 2\mathbb{E}[\mathbb{E}[y \mid f(x)]^2] + 2\mathbb{E}[\mathbb{E}[y \mid f(x)]y]]$$

$$= \mathbb{E}[y^2] + 2\mathbb{E}[(f(x)\mathbb{E}[y \mid f(x)]] - 2\mathbb{E}[f(x)y] \tag{28}$$

$$- \mathbb{E}[\mathbb{E}[y \mid f(x)]^2]$$

$$= \mathbb{E}[(2f(x) - y - \mathbb{E}[y \mid f(x)])(\mathbb{E}[y \mid f(x)]) - y] \tag{29}$$

$$= \mathbb{E}[(f(x) - y)(\mathbb{E}[y \mid f(x)] - y)] \tag{30}$$

$$+ \mathbb{E}[(f(x) - \mathbb{E}[y \mid f(x)])(\mathbb{E}[y \mid f(x)] - y)].$$

By the law of total expectation, we will write the above as

$$\mathrm{MSE}(f) - \mathrm{CE}_2(f)^2 = \mathbb{E}[\mathbb{E}[(f(x) - y)(\mathbb{E}[y \mid f(x)] - y) \tag{31}$$

$$+ (f(x) - \mathbb{E}[y \mid f(x)])(\mathbb{E}[y \mid f(x)] - y) \mid f(x)]].$$

Focusing on the inner conditional expectation, we have that

$$\mathbb{E}[(f(x) - y)(\mathbb{E}[y \mid f(x)] - y) + (f(x) - \mathbb{E}[y \mid f(x)])(\mathbb{E}[y \mid f(x)] - y) \mid f(x)]$$

$$= \mathbb{E}[y \mid f(x)](f(x) - 1)(\mathbb{E}[y \mid f(x)] - 1) + (1 - \mathbb{E}[y \mid f(x)])f(x)\mathbb{E}[y \mid f(x)]$$

$$+ \mathbb{E}[y \mid f(x)](f(x) - \mathbb{E}[y \mid f(x)])(\mathbb{E}[y \mid f(x)] - 1)$$

$$+ (1 - \mathbb{E}[y \mid f(x)])(f(x) - \mathbb{E}[y \mid f(x)])\mathbb{E}[y \mid f(x)] \tag{32}$$

$$= (1 - \mathbb{E}[y \mid f(x)])\mathbb{E}[y \mid f(x)] \geq 0 \quad \forall f(x) \tag{33}$$

and therefore

$$\mathrm{MSE}(f) - \mathrm{CE}_2(f)^2 = \mathbb{E}[(1 - \mathbb{E}[y \mid f(x)])\mathbb{E}[y \mid f(x)]] \geq 0. \tag{34}$$

$$\square$$

The expectation in Equation 34 is over variances of Bernoulli random variables with probabilities $\mathbb{E}[y \mid f(x)]$.

## B    RESULTS

Table 2 summarizes the marginal $L_2$ ECE and accuracy for the two datasets across multiple architectures and training loss functions. The scatter plots in Figures 6 and 7 show the accuracy and both $L_1$ and $L_2$ ECE, for top-label and marginal calibration on CIFAR-10 and CIFAR-100, respectively. KDE-CRE is trained by directly minimizing the metric that is evaluated, e.g., in the first column we minimize marginal $L_1$ calibration error and in the last column we optimize the $L_2$ top label calibration error. Other methods do not have the flexibility of choosing the type of calibration and the $L_p$ norm.

Table 2: Accuracy and marginal $L_2$ ECE (%) computed with 15 bins for different approaches, trained from scratch (Pre T) and after temperature scaling (Post T).

| Loss | Metric | | CIFAR-10 ResNet | ResNet (SD) | Wide-ResNet | DenseNet | CIFAR-100 ResNet | ResNet (SD) | Wide-ResNet | DenseNet |
|---|---|---|---|---|---|---|---|---|---|---|
| CRE | ECE | Pre T | 0.020 | 0.009 | 0.007 | 0.008 | 0.002 | 0.002 | 0.001 | 0.001 |
| | | Post T (NLL) | 0.007 | 0.005 | 0.008 | 0.004 | 0.002 | 0.001 | 0.001 | 0.001 |
| | Acc | Pre T | 0.925 | 0.926 | 0.950 | 0.947 | 0.700 | 0.728 | 0.797 | 0.756 |
| | | Post T (NLL) | 0.927 | 0.925 | 0.950 | 0.947 | 0.700 | 0.729 | 0.794 | 0.758 |
| MMCE | ECE | Pre T | 0.009 | 0.015 | 0.009 | 0.004 | 0.003 | 0.001 | 0.003 | 0.003 |
| | | Post T (NLL) | 0.013 | 0.009 | 0.009 | 0.005 | 0.002 | 0.001 | 0.002 | 0.003 |
| | Acc | Pre T | 0.929 | 0.925 | 0.947 | 0.944 | 0.693 | 0.723 | 0.767 | 0.748 |
| | | Post T (NLL) | 0.926 | 0.926 | 0.949 | 0.945 | 0.691 | 0.722 | 0.770 | 0.743 |
| FL-53 | ECE | Pre T | 0.013 | 0.020 | 0.026 | 0.005 | 0.003 | 0.002 | 0.003 | 0.002 |
| | | Post T (NLL) | 0.008 | 0.009 | 0.022 | 0.004 | 0.002 | 0.002 | 0.002 | 0.001 |
| | Acc | Pre T | 0.922 | 0.920 | 0.936 | 0.948 | 0.695 | 0.711 | 0.760 | 0.752 |
| | | Post T (NLL) | 0.923 | 0.919 | 0.936 | 0.949 | 0.693 | 0.712 | 0.763 | 0.753 |
| $L_2$ KDE-CRE | ECE | Pre T | 0.010 | 0.015 | 0.007 | 0.008 | 0.002 | 0.002 | 0.001 | 0.001 |
| | | Post T (NLL) | 0.004 | 0.012 | 0.008 | 0.009 | 0.002 | 0.002 | 0.001 | 0.001 |
| | Acc | Pre T | 0.930 | 0.922 | 0.950 | 0.943 | 0.707 | 0.713 | 0.797 | 0.757 |
| | | Post T (NLL) | 0.930 | 0.921 | 0.950 | 0.944 | 0.707 | 0.717 | 0.794 | 0.755 |

## C    RELATIONSHIP BETWEEN THE BINNED ESTIMATOR AND THE KERNEL DENSITY ESTIMATOR

Figure 8 shows an example of the binned estimator in a three-class setting on CIFAR-10. The points are mostly concentrated at the edges of the histogram, as can be seen from Figure 8b. The surface of the corresponding Dirichlet KDE is given in 8c.

Figure 9 shows the relationship between the binned estimator and our KDE-ECE metric. The points represent a trained Resnet-56 model on a subset of three classes from CIFAR-10. In every row, a differnt number of points was used to estimate the KDE-ECE.

## D    EXPERIMENTS FOR REBUTTAL

### D.1    TRAINING TIME MEASUREMENTS

In Table 3 we summarize the running time per epoch for training with (KDE-CRE) and without (CRE) regularization for the two datasets and four architectures. KDE-CRE does not create an overhead of more than a couple percent over the CRE baseline.

### D.2    CANONICAL CALIBRATION IN A MEDICAL APPLICATION

An additional experiment with a medical application, where the canonical calibration is of particular interest, was performed on the publicly-available Kather dataset (Kather et al., 2016), which consists of 5000 histological images of human colorectal cancer. The data has eight different classes of tissue. Figure 10 shows a comparison in performance of the CRE baseline with our KDE-CRE method. The canonical $L_1$ (left) and $L_2$ (right) calibration is measured using our KDE-ECE metric. The results clearly illustrate that our method significantly outperforms the cross-entropy baseline, both in terms of accuracy and calibration error, for several choices of the regularization parameter.

### D.3    BIAS AND CONVERGENCE RATES

Figure 11 shows a comparison of the groud truth, computed from 3000 test points with KDE-ECE against KDE-ECE and binned ECE estimated with a varying number of points used for the estima-

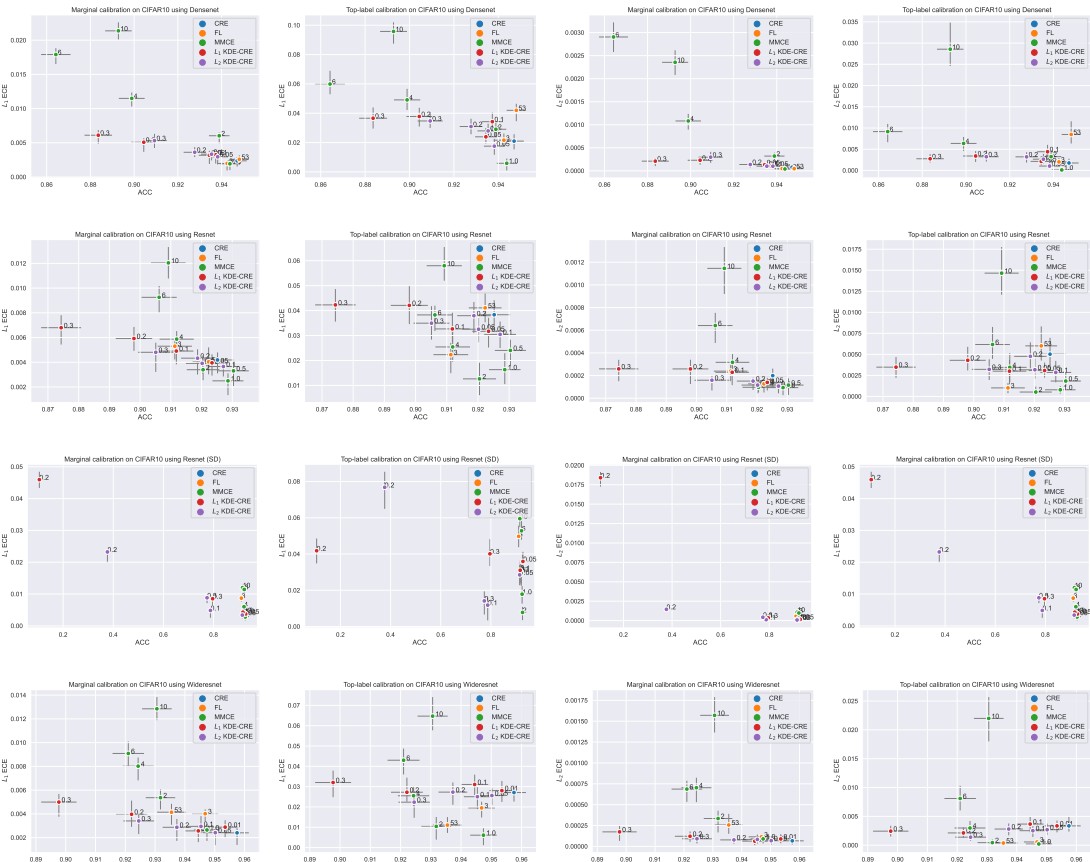

Figure 6: Top-label and marginal calibration on CIFAR-10.

Table 3: Training time [sec] per epoch for Cross-Entropy and KDE-CE methods for different models and datasets.

| Dataset | Model | CRE | $L_1$ KDE-CRE |
|---|---|---|---|
| CIFAR-10 | ResNet-110 | 51.8 | 53 |
| | ResNet-110 (SD) | 45 | 46 |
| | Wide-ResNet-28-10 | 152.9 | 154.9 |
| | DenseNet-40 | 103.2 | 106.8 |
| CIFAR-100 | ResNet-110 | 90 | 92.9 |
| | ResNet-110 (SD) | 78.2 | 80.7 |
| | Wide-ResNet-28-10 | 150.5 | 155.3 |
| | DenseNet-40 | 101 | 105.5 |

tion. The used model is a ResNet-56, trained on a subset of three classes from CIFAR-10. The figure shows that the two estimates are comparable and both are doing a reasonable job.

Figure 12 shows the absolute difference between the ground truth and estimated ECE using our KDE estimator and a binned estimator with varying number of points used for estimation. The results are

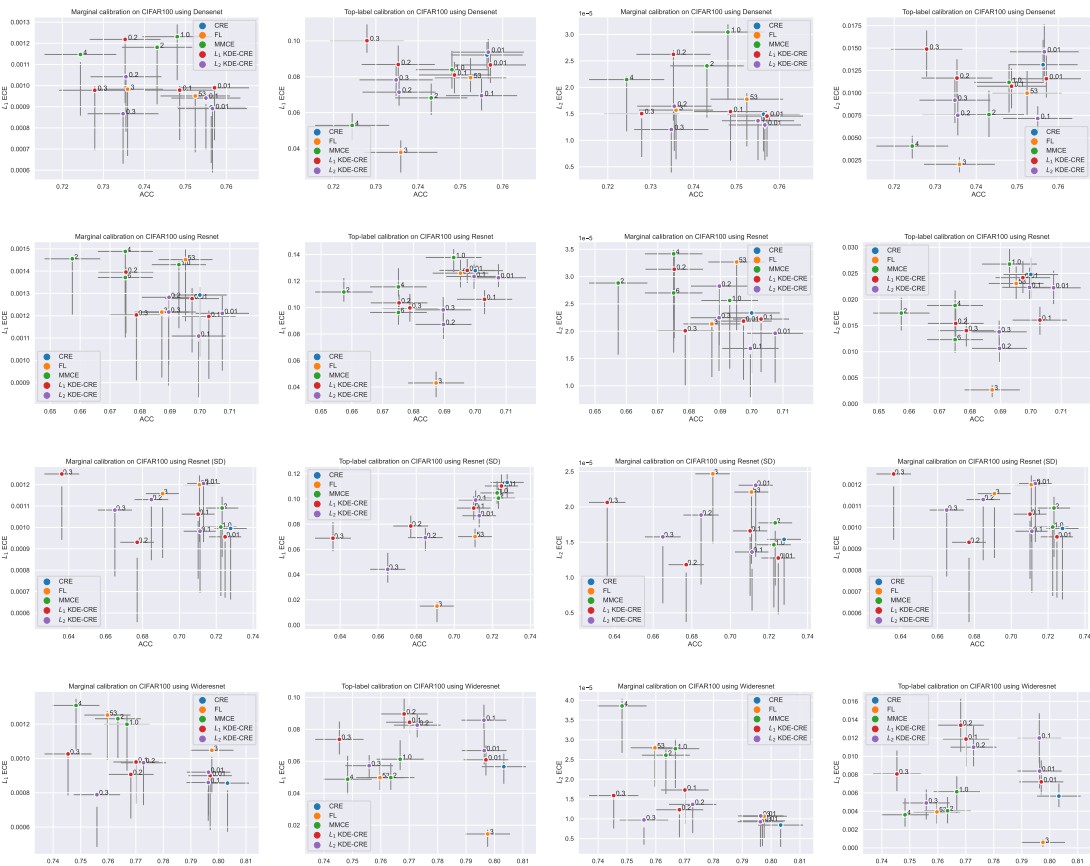

Figure 7: Top-label and marginal calibration on CIFAR-100

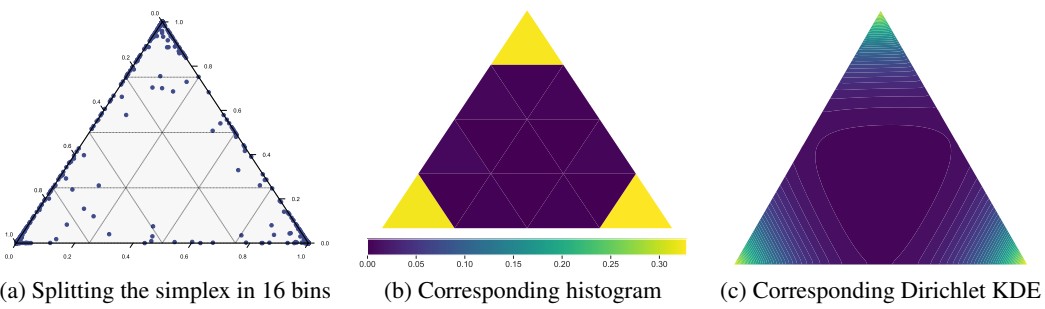

(a) Splitting the simplex in 16 bins      (b) Corresponding histogram      (c) Corresponding Dirichlet KDE

Figure 8: An example of a simplex binned estimator and kernel-density estimator for CIFAR-10

averaged over 120 ResNet-56 models trained on a subset of three classes from CIFAR-10. Both estimators are biased and have some variance, and the plot shows that the combination of the two is in the same order of magnitude. The empirical convergence rates (slope of the log-log plot) is given in the legend and is shown to be close to the theoretically expected value of -0.5.

### D.4 CHOICE OF THE BATCH SIZE

In Figure 13 we investigate the choice of the batch size on CIFAR-10. To this end, we use two differently shuffled dataloaders that draw random batches from the same training set. The first dataloader provides batches to the loss term (CRE) while the second dataloader provides the batches for the regularization (KDE). The batch size for the loss term is fixed in all experiments, while the

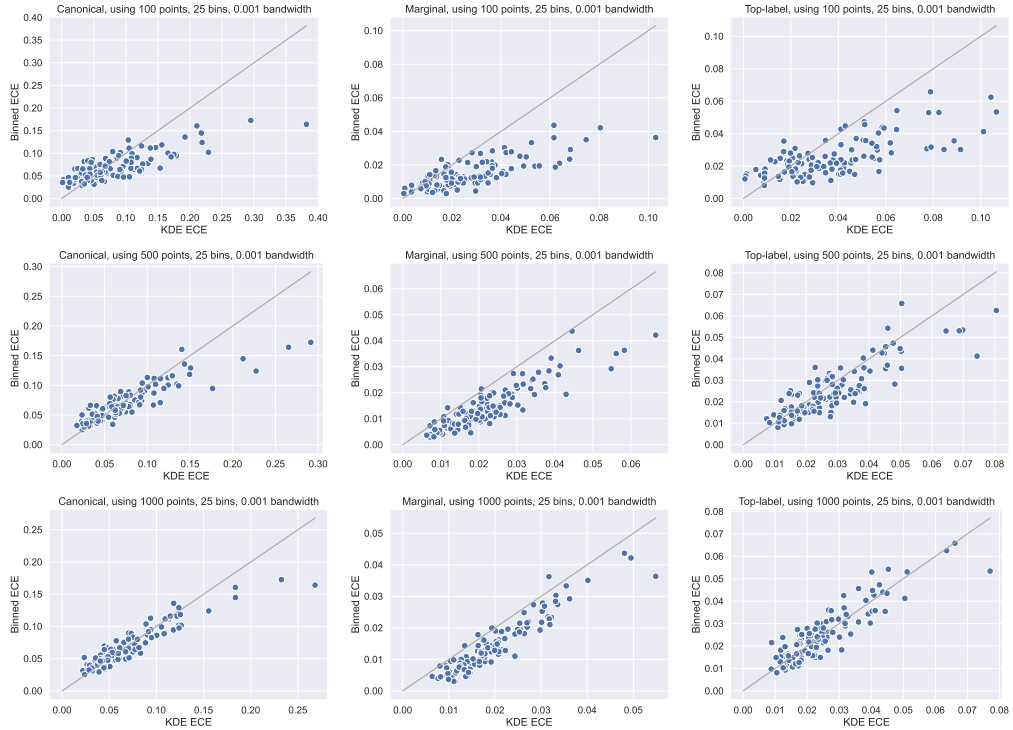

Figure 9: Relationship between the ECE metric based on binning and kernel density estimation (KDE-ECE) for the three types of calibration: canonical, marginal and top-label. In every row, a different number of points are used to approximate the KDE-ECE.

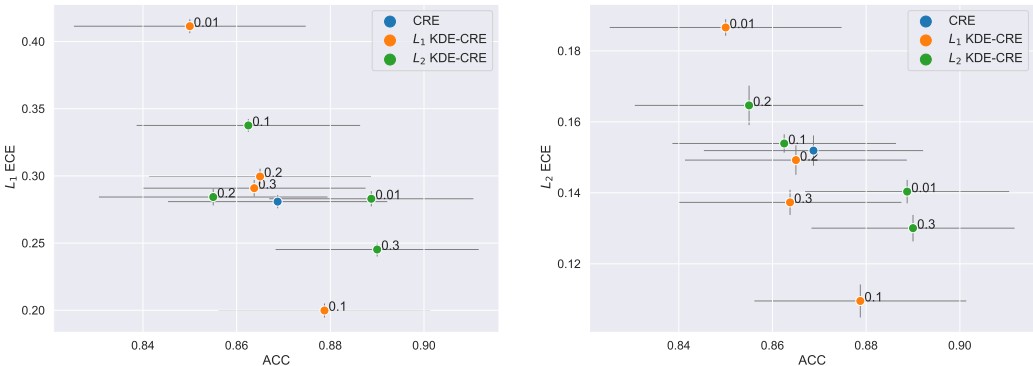

Figure 10: Canonical calibration on Kather using a Resnet-50 model

batch size for the regularization varies. The orange point is our normal experimental set-up with just one dataloader (i.e. the same points are used for loss and KDE-ECE computation) as a comparison. The plot shows that our chosen batch size of 128 is appropriate for our purposes.

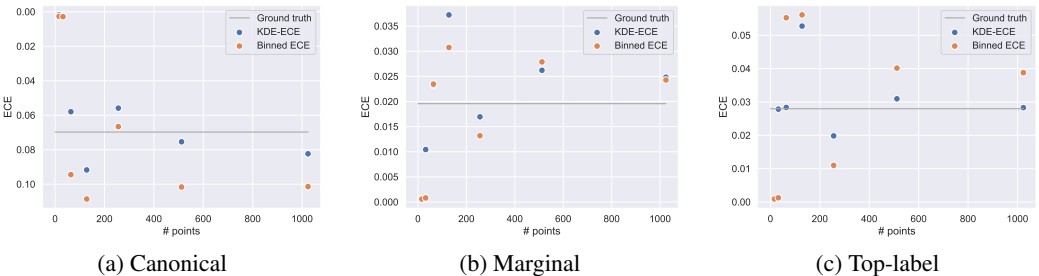

Figure 11: KDE-ECE estimates and their corresponding binned approximations on the three types of calibration for varying number of points used for the estimation. The ground truth is calculated using 3000 probability scores of the test set. For the binned estimate, the points are assigned in 25 bins with adaptive width. A bandwidth of 0.001 is used for KDE-ECE.

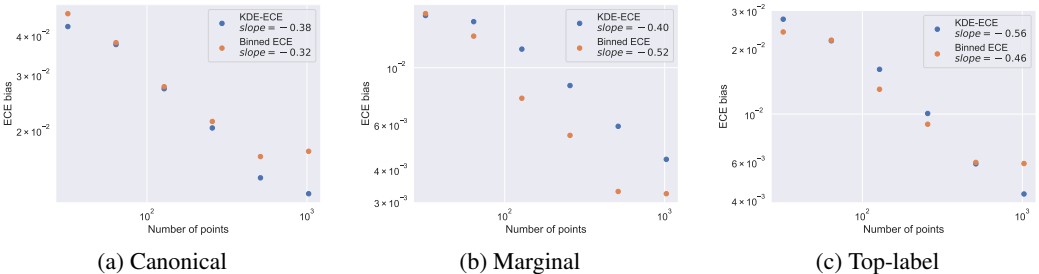

Figure 12: Absolute difference between ground truth and estimated ECE for varying number of points used for the estimation. The ground truth is calculated using 3000 probability scores of the test set. For the binned estimate, the points are assigned in 25 bins with adaptive width. A bandwidth of 0.001 is used for KDE-ECE. Note that the axes are on a log scale.

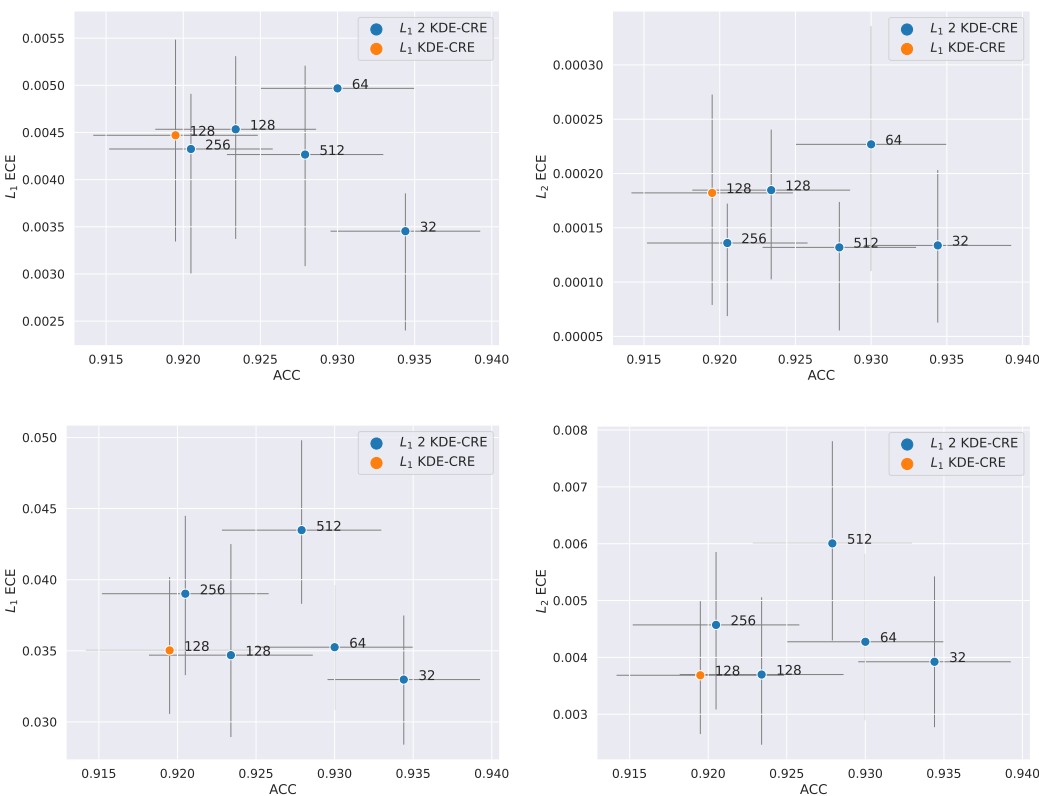

Figure 13: Training with different batches for loss and regularization (2 KDE-CRE), where the batch size for the loss is fixed and the batch size for the regularization varies. The orange point shows our usual experimental set-up where we train with only one batch (KDE-CRE). Upper row: marginal, lower row: top-label.

