# OpenReview forum: "Calibration Regularized Training of Deep Neural Networks using Kernel Density Estimation"
_ICLR.cc/2022/Conference — ICLR 2022 Submitted_

### Official Review · Reviewer_6Gih · 2021-11-01

**Correctness:** 3
**Technical Novelty And Significance:** 2
**Empirical Novelty And Significance:** 2
**Recommendation:** 5
**Confidence:** 5

**Main Review:**

This is an easy-to-follow paper and the proposed kernel estimator is simple to implement. The main idea is not very technically novel, but the introduction of beta/dirichlet kernels for classification is interesting, since they are differentiable and fit into the probability simplex.

Main concern:
The idea of differentiable ECE is not entirely new. Although histogram ECE are non-differentiable, some work have proposed remedy for this issue [1], and the kernel density ECE proposed in [2] also requires trivial change to be differentiable (has been explored by [3]). Considering this, I recommend the authors to discuss and compare with those approaches in this work.

As for the experimental results, I'm also not convinced that the proposed KDE-regularized training significantly outperformed the cross-entropy baseline (Table 1-2), as it seems that on many NN architectures the CE one obtained comparable or even better ECE. A major contribution of this paper is the introduction of Dirichlet kernels, but the authors missed an opportunity to demonstrate the benefit of using Dirichlet kernels over existing kernel density ECE method [2,3] or advanced histogram estimators (such as [4]) in the experimental section, such as their better capability to handle the multi-class cases, other than showing that they are "closely correspond to" the histogram ones. I believe this can be improved.

1. Bohdal O, Yang Y, Hospedales T. Meta-Calibration: Meta-Learning of Model Calibration Using Differentiable Expected Calibration Error[J]. arXiv preprint arXiv:2106.09613, 2021.

2. Zhang, Jize, Bhavya Kailkhura, and T. Yong-Jin Han. "Mix-n-match: Ensemble and compositional methods for uncertainty calibration in deep learning." International Conference on Machine Learning. PMLR, 2020.

3. Ma, Chunwei, Ziyun Huang, Jiayi Xian, Mingchen Gao, and Jinhui Xu. "Improving Uncertainty Calibration of Deep Neural Networks via Truth Discovery and Geometric Optimization." arXiv preprint arXiv:2106.14662 (2021).

4. Kumar, Ananya, Percy Liang, and Tengyu Ma. "Verified uncertainty calibration." arXiv preprint arXiv:1909.10155 (2019).

**Summary Of The Paper:**

In this paper, the authors proposed  the Beta/Dirichlet kernel density estimator, which is differentiable (trainable) and asymptotically converges to the true Lp calibration error. Empirical results demonstrated that such estimator is beneficial for both training and evaluation tasks.

**Summary Of The Review:**

Overall, I think this is an interesting paper, but not ready to publish in ICLR in its present form. I hope that the authors can improve the empirical aspects for this paper.

---

> ### Author Response · Authors · 2021-11-15
> **Initial response to Reviewer 6Gih**
>
> As an initial response, we would like to thank the reviewer for the interesting references, which we will be happy to cite in the final version of the paper. However, we would like to point out that two of them ([1], [3]) were uploaded to arXiv after June 5, and are therefore not considered to be prior work for ICLR submissions according to the ICLR reviewer policy: "if a paper was published (i.e., at a peer-reviewed venue) on or after June 5, 2021, authors are not required to compare their own work to that paper." As for the comparison with the other two papers, we discuss [2] in the Related work section and point out that the triweight kernel that the authors use does not have a natural extension to the multiclass case as the Dirichlet kernel and therefore we do not consider it in the experiments. Moreover, all the results in the paper considering the binned estimate are calculated with the procedure described in [4].
>
> We will address the other concerns shortly.

---

> > ### Author Response · Authors · 2021-11-22
> > **Response to Reviewer 6Gih**
> >
> > We thank the reviewer for the feedback and the interesting references. We are glad you found our paper easy-to-follow and simple to implement. We address your concerns below.
> >
> > **Concern that the idea of differentiable ECE is not entirely new.**
> >
> > It is true that other people have addressed this problem before, which implies that it is of strong interest in the ML community. As you have pointed out, the most closely related approach is the ICML 2020 paper [2], which also uses kernel density estimation. However it is important to note that their choice of kernel does not extend to canonical calibration as ours does, therefore it is not possible to compare in one of the main settings of interest in our paper. Moreover, they did not figure out that one actually has to use a ratio of KDEs. Finally, for the multiclass setting, they had to use a heuristic with numerical integration instead of doing it in $O(n^2)$ time for all dimensions. These are the reasons why we didn’t include this method in the experimental section, although we did address it in the related work section and mentioned its shortcomings. Evidently, getting the right kernel over the full simplex is a non-trivial task and we believe that the fact that we are the first to derive it is an excellent argument for accepting the paper.
> >
> > **KDE-CRE vs CRE**
> >
> > We kindly refer you to the second paragraph of our general response. In short, even though CRE performs well in some cases, our method is either on par or better than this baseline. As an additional experiment comparing the two, please see the revised version of the paper and our response to reviewer qGbY.

---

### Official Review · Reviewer_HhAZ · 2021-11-02

**Correctness:** 3
**Technical Novelty And Significance:** 3
**Empirical Novelty And Significance:** 2
**Recommendation:** 5
**Confidence:** 4

**Main Review:**

I found parts of this paper, particularly the exposition to be quite clear.  I also find the proposed method to be appealing in its simplicity and how easy it would be to apply to generic classification tasks (e.g., regardless of architecture).  I do find the results to be a bit lackluster (improvements seem marginal compared to existing methods) and some of the results are a bit cursory (e.g., the authors mention the application to any method that uses stochastic gradient descent, but the penalty requires looking at a large number of data points to estimate the density, and there is no guidance on how to set batch size or whether to use the whole dataset.  Does using smaller batches incur more bias? Is the bias large enough to be a concern for reasonable sample sizes?  The authors leave this for future work.) Otherwise, I have a number of mostly minor comments below.

Minor Comments:
- It would have been nice to go into depth about how to choose the various hyperparameters introduced by this method.  The authors describe a method for choosing the kernel bandwidth, but do not discuss how to choose $p$ in the $MCE_p$ regularization term, or how to set $\lambda$.  I suppose $p$ should be chosen based on which canonical calibration you want, and $\lambda$ might be chosen using cross-validation, but it would be good to provide users some guidance.
- Given that histogram density estimators are also consistent estimators of densities (given that the number of bins grows as sample size grows) Figures like Figure 2, Figure 3, Figure 8 and Figure 9 are statements about whether the Dirichlet KDE or the histogram is a more sensible estimator for a particular finite sample.  While it seems totally reasonable that the Dirichlet KDE should do much better (it's visually striking in Figures 2 and 8), Figures 3 and 9 are a bit difficult to interpret.  It's clear that the two estimates are highly correlated, but it's not obvious that one is better than the other.  One easy way to visualize this, would be to compare the estimates from using different numbers of points (as in Figure 9) but compare large samples to small samples: given the consistency of the KDE ECE, one could take the KDE ECE from 1000 points are being close to the ground truth, and then by plotting either the Binned ECE from 100 points or the KDE ECE using 100 points against this "ground truth" and see which estimator is better able to match the large sample size result with a smaller sample size.
- From a writing standpoint, I did find the overall organization of the paper to be a bit lacking which made it hard to follow the main thread -- for instance section 5.2 is interesting, and a nice use case of the Dirichlet KDE, but then it's not mentioned or used again (e.g., in the Tables etc...) and does not really belong in the "Results and Discussions" section.  Section 5.2 also serves to separate the details of the empirical exploration, from the results, making the paper feel disjointed.  Lastly, the "Mean squared error in binary classification" section is nice, but given that some "tricks" are used to reformulate the optimization problem, it feels a bit disconnected from the framework presented at the beginning of Section 3.

Typos:
- I believe that in the top paragraph on p. 6, $f(x) \in [0, 0.2)$ should be $f(y | x) \in [0, 0.2)$ of perhaps $f( \cdot | x) \in [0, 0.2)$.
- The abbreviation "ECE" is used in the last paragraph on p. 2, but is not defined until the section "Metrics" on p. 6.

**Summary Of The Paper:**

In this paper the authors address the problem of calibration in classification tasks, proposing to use kernel density estimation methods to estimate calibration, and then add a penalty during model training to regularize the model toward having better calibration.  They apply this technique to CIFAR-10 and CIFAR-100 and consider several measures of calibration.

**Summary Of The Review:**

The paper presents a conceptually simple approach to improving the calibration of classification models.  The paper is interesting, but could use some restructuring.  It is not totally clear how to use the method in practice (especially on large datasets where $O(N^2)$ would be prohibitive) and it is not clear if the proposed methodology substantially improves calibration over existing approaches.

---

> ### Author Response · Authors · 2021-11-22
> **Response to reviewer HhaZ (part 1)**
>
> We thank the reviewer for the detailed and thoughtful feedback. We are pleased that you appreciate the elegance of our solution. We address your questions and concerns below.
>
> **Improvements compared to existing methods**
>
> It is true that our method performs on par or better than existing methods, however it is important to note that our method extends appropriately to the highest form of calibration - canonical calibration - and thus enables full training for that type of calibration for the first time. Moreover, we have a consistent estimator amenable to high dimensional data and standard statistical debiasing techniques, which can lead to proper uncertainty estimates and hypothesis tests in future work.
>
> **The penalty requires looking at a large number of data points to estimate the density, and there is no guidance on how to set batch size**
>
> We are grateful that the reviewer has made us aware that we have not specified this in the paper and have done so in the revised version. We compute the kernel density estimate on a mini-batch, not on the whole dataset and therefore the computational overhead of our method is of no concern. We included a table (Table 3) of the running times per epoch with and without the regularizer in the revision.  It shows that the computational overhead of our method is indeed negligible for getting a calibrated model in return.
>
> Regarding the guidance about setting the batch size, we kindly refer the reviewer to Figure 13, where we keep the batch size for the loss term fixed and vary the batch size used for estimating the calibration error (we used two data loaders with different randomizations on the same training set). We see that there is a small variation in terms of accuracy and calibration error and we confirm that the standard values for batch size of 128 (Resnet, Wideresnet) and 64 (Densenet) are reasonable.
>
>
> **Bias vs batch size and KDE-ECE vs Binned ECE**
>
> We address here the comments on bias in the main review, as well as the second minor comment as we believe these points are strongly related.
>
> Thank you for the insightful questions regarding the effect of the bias on the batch size and the concrete suggestion for plotting the Binned ECE and KDE ECE. Per your suggestion, we included Figure 11 in the revision where the “ground truth” is obtained from KDE ECE using 3000 points and we plot the Binned ECE and the KDE-ECE estimated using a varying number of points (32, 64, ...,1024). The figure shows that the two estimates are comparable and both are doing a reasonable job.
>
> Additionally, we added Figure 12 which measures the absolute difference between ground truth and estimated ECE, averaged over 120 models trained on subsets of 3 classes from CIFAR-10. Both estimators are biased and have some variance, and the plot shows that both are of similar magnitude and have similar convergence rate.
>
> In conclusion, the binned estimator is the most commonly used and accepted metric for evaluating calibration error, in spite of its limitations. We have shown empirically and theoretically (see response to qGbY regarding the convergence rate) that the KDE-ECE has very similar statistical properties in terms of bias and convergence. On the other hand, the advantage of using KDE-ECE over the binned estimate (discussed in section 5.2) can be summarized as follows: KDE-ECE is trainable, it converges as fast as the binned estimate, the error terms are in the same order of magnitude as the binned estimate, it extends to canonical calibration and has better scaling properties with respect to the output dimension.
>
> **Choosing hyperparameters**
>
> We appreciate your comment and have realized that the discussion for choosing the hyperparameters was scattered in different sections of the paper. Therefore, we moved it to a new paragraph under the Empirical setup section. It explicitly says that indeed the choice of p should be based on the desired type of Lp calibration (the choice might be related to what type of theoretical guarantees you want as different guarantees have been proven using different values of p in the literature) and the lambda can be chosen using cross validation or using a holdout validation set. In practice, we expect that $p\in \{1,2,\infty\}$ are as usual the most likely choices, where $p=\infty$ can be approximated e.g. by some finite $\tilde{p}\geq 4$.  We expect that our paper and accompanying software release will accelerate community research on this question.

---

> > ### Author Response · Authors · 2021-11-22
> > **Response to reviewer HhaZ (part 2)**
> >
> > **Structure of the paper**
> >
> > We thank the reviewer for their input and suggestions regarding the structure of our paper.
> >
> > Regarding section 5.2, one of the key advantages of our method is the ability to naturally extend to a probability simplex, thus enabling estimating canonical calibration. We believe that this section is crucial for empirically presenting our estimator and comparing it to the widely adopted binned estimator. Furthermore, this estimator of canonical calibration is used in the following section 5.3 (Figure 4), which illustrates the performance of the multiple architectures on CIFAR-10 in terms of accuracy and canonical calibration.
> >
> > Regarding the section on mean squared error in binary classification, we believe that the derivation of the sharpness gives a nice warm-up to the derivation of the full term and think that this will help the reader to understand the general solution by working out in detail one specific example. Moreover, it helps to focus on the sharpness term which we can partially debias, thereby mapping the subsequent statistical analysis of our estimator. That in turn is related to our above discussion about convergence.
> >
> > **Typos**
> >
> > Thanks for pointing out the typos, we have corrected this in the revised paper.

---

### Official Review · Reviewer_JEjT · 2021-11-02

**Correctness:** 4
**Technical Novelty And Significance:** 3
**Empirical Novelty And Significance:** 2
**Recommendation:** 5
**Confidence:** 3

**Main Review:**

Strengths:
-  Idea is conceptually simple
- The work focuses on the stronger notion of calibration (canonical)

Weaknesses:
- The method claims to be well suited for differentiable training. Yet, these are mostly deep learning scenarios with large datasets. The scalability of the O(n^2) estimator can be a challenge. The paper does not provide any data on training times. A breakdown of "training without CE term" vs "training with CE term" would be very helpful in assessing the utility of this method, especially in evaluating whether the improvement in calibration error relative to other methods is worth the additional computational budget.

**Summary Of The Paper:**

The paper proposes a regularization term to augment loss functions, where the regularization term effectively minimizes the calibration error of the model. The term itself is a kernel density estimator over the K-simplex space (hence Dirichlet kernel is the natural choice).

The authors claim the estimator is consistent (but not unbiased, though they partially debias it) and empirically verify that their method yields tradeoff between accuracy and calibration that is near Pareto optimal.

**Summary Of The Review:**

Justification of the recommendation is based on the strengths and weaknesses stated above.

---

> ### Author Response · Authors · 2021-11-15
> **Response to Reviewer JEjT**
>
> We thank the reviewer for the valuable feedback and we appreciate the opportunity to clear out an important fact regarding the complexity of O(n^2), which was raised also by other reviewers.
> We would like to first point out that in our experiments, the density is estimated using a mini-batch, so the complexity of O(n^2) is w.r.t. the mini-batch and not the entire dataset. Considering commonly used batch sizes (in our experiments, 64 for DenseNet and 128 for the other architectures), the computational overhead is not concerning.
> Furthermore, we would like to emphasize that the complexity of the alternative trainable approach for calibration regularized training (MMCE) [1] also has a complexity of O(n^2), as does the other KDE-based ECE estimator [2], which relies on numerical integration.
> We apologize for not being clear enough in the original version that the complexity O(n^2) is w.r.t. the mini-batch and we will correct this in the revision.
>
> We appreciate the suggestion to compare the training times with and without the CE term and we will include a table in the revised version of the paper. The breakdown of training with and without the CE term is already provided in Table 1 in terms of accuracy and ECE (last and first row, respectively).
>
> [1] Aviral Kumar, Sunita Sarawagi, and Ujjwal Jain. "Trainable calibration measures for neural networks from kernel mean embeddings." ICML 2018.
>
> [2] Zhang, Jize, Bhavya Kailkhura, and T. Yong-Jin Han. "Mix-n-match: Ensemble and compositional methods for uncertainty calibration in deep learning." International Conference on Machine Learning. PMLR, 2020.

---

> > ### Author Response · Authors · 2021-11-22
> > **Additional response to Reviewer JEjT**
> >
> > We kindly refer you to our general response where we address the concern about the $O(n^2)$ complexity. Briefly, we believe that the complexity of $O(n^2)$ is intrinsic to the problem of KDE and shows up in other works [1, 2]. On the other hand, it appears that in order to get significant gains it’s sufficient to estimate it with a minibatch.
> >
> > We compared the training times per epoch (Table 3) and found that the added cost per epoch of our method is a few seconds, resulting in only a couple percent overhead.

---

### Official Review · Reviewer_qGbY · 2021-11-04

**Correctness:** 3
**Technical Novelty And Significance:** 3
**Empirical Novelty And Significance:** 3
**Recommendation:** 8
**Confidence:** 3

**Main Review:**

Strengths:

1. A novel and technically sound approach for regularizing neural networks in terms of the canonical calibration error is proposed with competitive empirical performance.

2. The use of the Dirchlet Kernel Density Estimate to measure the calibration error is an interesting side benefit of the approach.

Weaknesses:

1. The technical novelty is a bit limited since it mainly consists of directly plugging in the expression for the Dirichlet/Beta Kernels into the regularized loss functions. The consistency results are a good starting point but without a discussion on convergence rates they are not very useful in practice.

2. The  approach seems a bit computationally expensive especially since the simple CRE approach has competitive marginal L1 ECE performance (both pre and post T) and is much less expensive. I would suggest either exploring a more computationally efficient version of the approach (For eg. using approximations to speed up the kernel computations) or including an application (in experiments) where the canonical L1 ECE is more important (such as the medical diagnosis application mentioned in the introduction) so as to clearly illustrate a setting where the proposed approach would be preferrable to CRE. Also given the difference in computation cost I would like to see inference time measurements to exactly illustrate the overhead w.r.t the CRE baseline.

Additional questions:

1. Is it possible to consider f-divergences instead of the Lp error in CE under your approach? If yes, have you already looked into it?

2. I think there is some issue with the subscripts in (19). If $z_i$ is a softmax output (vector) then shouldn't the individula $\alpha$'s be labeled as $alpha_{ij}$?

3. Why are values of $\lambda$ different fro MMCE and KDE-CRE? Does the $\lambda$ in MMCE correspond to the value used in the original paper?

**Summary Of The Paper:**

The paper proposes a new approach for calibrating neural network outputs. The idea is to train the neural network with a regularized loss function that is a linear combination of prediction and calibration errors. The calibration error is measured as the Lp norm of the difference between predicted class probabilities and the expected true class probabilities given the predicted class probabilities where the latter term is computed using a kernel density estimate with a Dirichlet Kernel. The approach is evaluated in terms of accuracy and expected calibration error on CIFAR-10 and CIFAR-100.

**Summary Of The Review:**

I am leaning towards accepting the paper because the approach is technically sound and the results are competitive. However I would like to see more of a discussion on the comparison of the proposed approach with the less expensive CRE baseline (see my recommendations above) to be completely convinced.

Comments after rebuttal: I am satisfied by the additions made to the paper. The experiments on the Kather (medical) dataset do illustrate the superior L1 ECE (canonical) of the proposed approach, and the time measurements show that the overhead introduced by the proposed approach is not significant. I am therefore increasing my score to reflect the same.

---

> ### Author Response · Authors · 2021-11-22
> **Response to Reviewer qGbY**
>
> We thank the reviewer for the valuable feedback. We are encouraged that you found our approach novel, technically sound, interesting and with competitive empirical performance. We address your concerns below.
>
> **Technical novelty**
>
> We respectfully object that the technical novelty of our approach is limited and only consists of plugging in the Dirichlet kernel. First we would like to point out that many people have been working in this direction, but have not derived the correct solution. For instance, the most closely related paper (Mix-n-Match [2] - published at ICML 2020) does not estimate canonical calibration. They also did not figure out that one needs to take a ratio of density estimators, which we derived. Furthermore, they did not figure out that you could have a solution that is $O(n^2)$ for all dimensions and therefore they had to rely on numerical integration for the multiclass setting. Evidently, getting the right kernel over the right space is not trivial.
>
> **Convergence rate**
>
> Regarding the convergence rate we can focus the discussion on the sharpness term since for the other terms of the $Lp$ calibration error the argument will be analogous. The sharpness term is estimated by a ratio of second order U-statistics that converge as $1/sqrt(n)$ (see e.g. Serfling (2009), Ch. 5, or Ferguson (2005) cited in the submission). Therefore, the overall convergence will be $1/sqrt(n)$. We kindly refer the reviewer to Figure 12 in the Appendix, which shows a log-log plot of the absolute difference between ground truth and estimated ECE using a varying number of points. The slope of the log-log plot, which corresponds to the empirical convergence rate, is shown to be close to the theoretically expected value of -0.5.
>
> **Computational cost**
>
> For your concerns w.r.t. the computational cost of our method we kindly refer you to our general response. In short, we estimate the density on mini-batches and hence the $O(n^2)$ dependence is of no concern in practice. Furthermore, this matches the computational complexity of other works [1, 2] and we believe that for density estimates the $O(n^2)$ dependence is intrinsic to the problem.
>
> **Exploring a more computationally efficient version of the approach**
>
> Your suggestion to use approximations to speed up the kernel computations is indeed a very interesting future research direction, in particular if one wants to evaluate the calibration error for large datasets. However, for calibration regularized training we found that an estimate per batch is sufficient and the overhead of the computational cost (see Table 3) is very minimal and certainly worth the calibrated model that we obtain in turn. Moreover, as we have already stated in the paper, one of our follow up works will be to come up with a debiasing procedure for our estimator which then will allow us to use a block estimator to speed up the computations (e.g. analogous to the approach of Zaremba et al. (NeurIPS 2013) in the context of MMD, which was used by [1]).
>
> **Include an application where the canonical L1 ECE is more important (e.g. medical diagnosis application), to illustrate a setting where the proposed approach is preferable to CRE.**
>
> Thank you for the concrete suggestion. We included an additional experiment on a medical dataset consisting of histological images of human colorectal cancer. Per your request, we measure canonical ECE (both L1 and L2) and compare it with the CRE baseline. We kindly point you to Figure 10 in our revised paper, which clearly illustrates our method significantly outperforms the cross-entropy baseline, both in terms of accuracy and calibration error,  for several choices of the regularization parameter.
>
>
> **Inference time measurements**
>
> We fully agree that the time measurements are a valuable addition to our paper and we are grateful for your helping us to improve it. We included Table 3 with training time measurements per epoch. As previously stated, the overhead is typically a couple percent as it adds a few seconds of extra computation, which is negligible considering the fact we are getting a better calibrated model.
>
>
> **F-divergences**
>
> We agree that f-divergences are another natural way to measure miscalibration. We believe that Dirichlet KDEs are a promising strategy for f-divergences as well since in our code we work with the Dirichlet kernel in log space which is very similar to KL divergence computations.  This is an interesting area of future research.
>
> **Typo in (19)**
>
> Thank you!
>
> **Lambda**
>
> The values for lambda are different for MMCE and KDE-CRE because the objective functions are the result of different Lagrangians and therefore the trade-off between the terms is different. The lambda in MMCE corresponds to optimal values as reported in the original paper.

---

> > ### Comment · Reviewer_qGbY · 2021-11-26
> > **Response to rebuttal**
> >
> > I am satisfied by the additions made to the paper. The experiments on the Kather (medical) dataset do illustrate the superior L1 ECE (canonical) of the proposed approach, and the time measurements show that the overhead introduced by the proposed approach is not significant. I am therefore increasing my score to reflect the same.

---

### Author Response · Authors · 2021-11-22
**General response**

We thank all the reviewers for their time and efforts in reviewing our paper. We appreciate the insightful feedback and constructive suggestions. We have updated the paper to address the raised concerns.

In summary, one primary concern shared by multiple reviewers (**qGbY, JEjT, HhAZ**) is the $O(n^2)$ complexity of our method. We are thankful to the reviewers for bringing to our attention the fact that we have not clearly stated in the paper that within the proposed calibration-regularized training framework, the density is estimated using a mini-batch, not the entire dataset, and therefore the overhead is negligible. More concretely, per the request of multiple reviewers, we included Table 3 in the Appendix that shows time measurements comparing the training time per epoch of a method with (KDE-CRE) and without regularization (CRE). The added cost of our method is a few seconds (typically a couple percent overhead).
Finally, we’d like to point out that the complexity of $O(n^2)$ shows up in other related works [1, 2] and we believe it is intrinsic to the problem of density estimators of calibration error, rather than a shortcoming of our method. On the other hand, our empirical results show that in order to obtain a significant reduction in calibration error it’s sufficient to estimate the density on a mini-batch with commonly used batch sizes.

Another shared concern of reviewers **qGbY** and **6Gih** is with regard to the good performance of the cross-entropy baseline in our experiments. It is true that our empirical results are on par or better (e.g. Resnet 110 and Densenet-40) than the cross-entropy baseline, however the fact that sometimes cross-entropy performs well should not be an argument against our method, which improves the calibration error in a principled way in cases where cross-entropy fails. Per the suggestion of reviewer **qGbY**, we added an additional experiment with a medical application where the canonical calibration is of particular interest. The results on Figure 12 clearly illustrate that our method significantly outperforms the cross-entropy baseline, both in terms of accuracy and calibration error, for several choices of the regularization parameter.

We address the individual concerns and questions below.

List of revisions in the new manuscript (changes in the main text are highlighted in red in the revised paper):
- We added a table of the time measurements per epoch comparing the training with and without calibration regularization (Table 3 in Appendix)
- We added an additional experiment on medical data, comparing the CRE baseline with our approach (Figure 10 in Appendix)
-We added a plot comparing “ground truth” ECE with the estimations from the KDE estimator and a binned estimator with varying number of points (Figure 11 in Appendix)
-We added a plot that compares the absolute difference between ground truth and estimated ECE for a varying number of points used for the estimation and showed the empirical convergence rates (Figure 12 in Appendix).
-Other minor revisions and suggestions pointed out by the reviewers were carried out

We believe that clearing out the confusion about the $O(n^2)$ complexity, adding the training measurements table that explicitly shows that the overhead is negligible and adding an additional experiment which showcases the superiority of our method against the CRE baseline have addressed the biggest concerns of the reviewers and improved the paper. We remain convinced that our approach is of strong interest to the ICLR community and has the potential for high impact.


&nbsp;&nbsp;[1] Aviral Kumar, Sunita Sarawagi, and Ujjwal Jain. "Trainable calibration measures for neural networks from kernel mean embeddings." ICML 2018.

&nbsp;&nbsp;[2] Zhang, Jize, Bhavya Kailkhura, and T. Yong-Jin Han. "Mix-n-match: Ensemble and compositional methods for uncertainty calibration in deep learning." International Conference on Machine Learning. PMLR, 2020.

---

### Decision · Program_Chairs · 2022-01-20

**Decision:**

Reject

**Comment:**

Thank you for your submission to ICLR.  The paper proposes a simple method for improving calibration performance using a loss based upon a Dirichlet KDE.  The method is appealing in its simplicity, but several reviewers (and myself) have concerns simply about the fact that the method ultimately seemed to give rather marginal improvement over the standard cross-entropy baseline.  The authors attempted to address this point in the rebuttal, with their additional example on the Kather domain.  And while this is a nice addition, I'm still not fully convinced that the improvement here is _that_ significant, to the point where I think it would be important to consider much broader sweeps of hyperparameters, etc, for all methods (which I believe should be reasonable here given the data set sizes).  I believe this has the potential to be a nice contribution, and its simplicity can be a positive, but ultimately I think a bit of additional effort is required to show the full empirical advantages of the method.